# Nipah and Hendra Viruses: Deadly Zoonotic Paramyxoviruses with the Potential to Cause the Next Pandemic

**DOI:** 10.3390/pathogens11121419

**Published:** 2022-11-25

**Authors:** Sabahat Gazal, Neelesh Sharma, Sundus Gazal, Mehak Tikoo, Deep Shikha, Gulzar Ahmed Badroo, Mohd Rashid, Sung-Jin Lee

**Affiliations:** 1Division of Veterinary Microbiology and Immunology, Faculty of Veterinary Science and Animal Husbandry, Sher-e-Kashmir University of Agricultural Sciences and Technology of Jammu, R.S. Pura, Jammu 181102, Jammu and Kashmir, India; 2Division of Veterinary Medicine, Faculty of Veterinary Science and Animal Husbandry, Sher-e-Kashmir University of Agricultural Sciences and Technology of Jammu, R.S. Pura, Jammu 181102, Jammu and Kashmir, India; 3Division of Veterinary Microbiology, College of Veterinary Sciences, Guru Angad Dev Veterinary and Animal Science University, Ludhiana 141004, Punjab, India; 4Department of Applied Animal Science, College of Animal Life Sciences, Kangwon National University, Chuncheon 24341, Republic of Korea

**Keywords:** Nipah virus, Hendra virus, BSL-4, vaccines, Cedar virus

## Abstract

Nipah and Hendra viruses are deadly zoonotic paramyxoviruses with a case fatality rate of upto 75%. The viruses belong to the genus henipavirus in the family *Paramyxoviridae*, a family of negative-sense single-stranded RNA viruses. The natural reservoirs of NiV and HeV are bats (flying foxes) in which the virus infection is asymptomatic. The intermediate hosts for NiV and HeV are swine and equine, respectively. In humans, NiV infections result in severe and often fatal respiratory and neurological manifestations. The Nipah virus was first identified in Malaysia and Singapore following an outbreak of encephalitis in pig farmers and subsequent outbreaks have been reported in Bangladesh and India almost every year. Due to its extreme pathogenicity, pandemic potential, and lack of established antiviral therapeutics and vaccines, research on henipaviruses is highly warranted so as to develop antivirals or vaccines that could aid in the prevention and control of future outbreaks.

## 1. Introduction

Nipah and Hendra viruses are highly virulent zoonotic paramyxoviruses which have been classified as BSL-4 (Biosafety Level-4) pathogens owing to their extreme pathogenicity and lack of established antiviral therapeutics and vaccines. Hendra virus was first isolatedfrom cases ofsevere respiratory and neurological disease in horses in 1994 in Brisbane, Australia, in turn resulting in the death of a human handling the affected equines [1]. Nipah virus was first recognized in an outbreak of respiratory and neurological disease in pigs and subsequent cases of encephalitis in pig farmers in Malaysia and Singapore in 1998–1999 [2]. Nipah and Hendra viruses are closely related and belong to the genus Henipavirus in the family *Paramyxoviridae*. Nipah virus infection can manifest as a wide array of clinical presentations ranging from asymptomatic to acute respiratory manifestation (particularly in India and Bangladesh) as well as fatal encephalitis (Malaysian outbreak). These viruses are deadly, with an estimated case fatality rate of 40% to 75% for the Nipah virus [3]. No vaccines or antivirals are licensed for use in humans against the Nipah and Hendra viruses, even though a vaccine commercialized under the trade name of Equivac has been licensed for use against the Hendra virus in horses. The absence of established antivirals and vaccines for use in humans coupled with high case fatality rate of these viruses make the Nipah and Hendra viruses a cause of concern. In fact, an urgent need for research on these viruses is highly warranted as per the 2018 annual review of the WHO R&D Blueprint list of priority diseases [3]. Considering all these factors, it is important to conduct research on the Nipah and Hendra viruses and attempt to develop antivirals or vaccines to control any future outbreaks. The review aimed to explain the details of Henipaviruses with special emphasis on the Nipah and Hendra viruses with respect to their current and previous outbreaks, transmission patterns, current diagnostic approaches, and recent experimental approaches for vaccination.

## 2. Henipaviruses: A Cause of Grave Concern

Henipaviruses, particularly the Nipah virus, have many characteristics which make them a significant threat to human and animal health. Firstly, fruit bats (*Pteropus* genus), which are natural reservoirs of the virus, are widely distributed throughout Asia, leading to frequent spillover events. Second, they can be transmitted from bats to humans directly or via domestic animals. Third, they are capable of human-to-human transmission. Fourth, spillover has frequently occurred in highly populated regions. Fifth, they have a high mortality rate in humans. Lastly, there are no vaccines to prevent or antivirals to mitigate the disease [4]. Thus, considering all of the abovementioned factors, NiV poses a significant threat to human health. In humans, NiV infection has been associated with a range of clinical manifestations, including widespread vasculitis, encephalitis, and myocarditis, with approximate mortality rates of 40% to 70% [5]. Infected individuals can sometimes exhibit symptoms such as drowsiness and mental confusion, which can progress to coma. Residual neurological complications have been reported in some survivors [6]. The clinical presentation of NiV infections in humans, swine, and equine is shown in Figure 1.

NiV was first identified in Malaysia and Singapore in 1998–1999 after an outbreak of respiratory and neurological disease in pigs, soon followed by cases of encephalitis in pig farmers. The virus isolated from the cerebrospinal fluid of patients that had died of the infection cross-reacted with HeV antibodies and was genetically closely related to HeV. The virus was named Nipah after the Malaysian village where one of the patients lived. The outbreak led to 105 human fatalities, with a total of 265 reported cases in Malaysia, and 1 human fatality with 11 cases reported in abattoir workers in Singapore. The disease had a huge economic impact, as 1.1 million swine had to be sacrificed to contain the infection [7]. In 2001, a different strain of NiV was identified as an etiological agent responsible for cases of encephalitis reported in Bangladesh, namely NiV-B. Since then, NiV infections in humans have been reported in Bangladesh almost annually. In fact, from 2001 to 2015, a total of 260 NiV cases have been confirmed in Bangladesh [8].

NiV outbreaks have also been reported in India, with the largest one being in Siliguri, West Bengal in 2001 (around 66 cases and 45 deaths) and a smaller outbreak with five cases and a 100% fatality rate reported in 2007 in Nadia district, West Bengal [9]. These outbreaks occurred across the border from the Nipah belt in Bangladesh; however, recently, in 2018, a NiV outbreak occurred in the Kozhikode and Malappuram districts of Kerala, a region in South India which is geographically disconnected from the areas affected by previous outbreaks [10]. In 2019, a single NiV case was identified in Kerala’s Ernakulum district. More recently, on 1 September 2021, NiV encephalitis was confirmed in a 12-year-old boy in Kozhikode, Kerala who succumbed to his illness on 5 September 2021 [6].

Cases of NiV encephalitis have also been reported in Philippines; in 2014, a total of 17 human cases were reported, out of which 11 exhibited symptoms of acute encephalitis while 6 cases exhibited influenza-like illness or meningitis.

In sum, a total of 650 human cases of NiV infection have occurred in Southeast Asia since 2001 with a combined case fatality rate of 60% [5].

Similar to NiV, the natural reservoirs of HeV are *Pteropus* bats, which result in fatal disease in both horses and humans. The virus can be transmitted to horses after exposure to excretions of HeV-infected bats, particularly urine. HeV can further be transmitted from horses to humans through excretions or the body fluids of horses infected with HeV. In humans, the symptoms of HeV infection can range from mild influenza-like symptoms to fatal respiratory disease or neurological manifestations. As with NiV, no vaccine or antivirals have been licensed for use against HeV in humans; however, a vaccine has been approved for use in horses (discussed in Vaccine section under subunit vaccines).

The first outbreak of the Hendra virus presented as a severe respiratory disease occurring in horses in 1994 in Brisbane (a suburb of Hendra), Australia, leading to the death of 14 horses and their trainer. Seven other horses and one person were also infected, but did not succumb to the infection. A novel paramyxovirus was identified as the etiological agent and was termed equine morbillivirus, which is now referred to as HeV [11,12,13]. However, the first known case of HeV occurred a few months prior when there were cases of HeV infection in horses; a person who assisted in the necropsy of the horses developed fatal encephalitis and died 13 months later [14]. Since the first identification of HeV in Australia, these viruses have emerged 62 times in Australia, resulting in 104 equine fatalities either due to fatal infection or euthanization [15]. In addition, a total of seven human cases and four deaths have been attributed to HeV with all human infections acquired via virus shed by horses [16].

Other than being fatal to humans and animals, henipavirus infection can also result in huge economic losses, which can be ascertained by the fact that 1.1 million swine had to be sacrificed to contain the 1998 NiV outbreak in Malaysia [7]. This resulted in a loss of US $97 million in compensation for the pigs that had to be culled to contain the outbreak. In addition, an indirect cost of US $229 million was incurred due to lost tax revenue to the government as well as losses in international trade. Furthermore, an additional US $136 million was lost for control programmes, viz. for biosecurity and animal slaughter [17]. The pork industry was severely affected by the Malaysian outbreak, which resulted in long-term effects post-outbreak as a result of altered pork consumption and pork export. The pork consumption and export dropped by around 80% during the Malaysian outbreak, while a dip of 30% was observed post-outbreak [18].

The importance of henipaviruses, especially NiV as a human pathogen, is further accentuated by the fact that NiV has crossed the species barrier on several occasions, resulting in infections in humans and animals with limited person-to-person transmission [19]. The transmission of NiV to humans occurs mainly in places where bats, pigs, and humans come in close proximity to one another. People rear pigs for economic benefits and in the vicinage of the farms, plant fruit-bearing trees for shade. Fruit bats belonging to the genus *Pteropus* are attracted to fruits and drop partially eaten saliva-laden fruits, which are consumed by domestic animals and thus result in spillover of infection to domestic animals/pigs from which transmission to humans can occur. This was particularly the case with the Malaysian outbreak. NiV outbreaks have also occurred by consumption of raw date palm sap (in Bangladesh), undercooked infected meat, and by handling infected animals [20]. Thus, NiV infection in humans can occur via intermediate hosts or directly from bats to humans and can also result in human-to-human transmission. Considering all these factors and due to the human-to human-transmission patterns for NiV infections coupled with high case fatality rates, NiV poses a significant threat to human health, particularly due to the lack of available vaccines and antivirals.

## 3. Transmission Cycle from Bats to Humans

The natural reservoir of NiV and HeV are fruit bats, such as flying foxes. The viruses are transmitted to humans mainly via intermediate hosts, viz. swine for NiV-M (Malaysian strain) and equines for HeV; however, direct bat-to-human and human-to-human transmission can also occur for NiV.

For the Hendra virus, the virus is transmitted from bats to horses (intermediate host) and from horses to humans coming in direct contact with secretions of infected equines. No human-to-human transmission has been documented yet for HeV. For Nipah, the transmission cycle of Malaysian strain (NiV-M) and Bangladesh strain (NiV-B) is distinct with the involvement of pigs in NiV-M, while transmission directly from bats to humans for NiV-B. In Malaysia, pig farms lie in close proximity to fruit trees, which are resided by fruit bats and domestic pigs contract NiV infection by coming into contact with materials contaminated by bat secretions, particularly by consumption of bat-eaten and saliva-laden fruits or by exposure to bat urine. NiV is subsequently transmitted from pigs to humans by direct contact. In India and Bangladesh, the major route of NiV transmission from bats to humans is by the consumption of raw date palm sap (a cultural delicacy in Bangladesh) contaminated with bat saliva or urine. However, human-to-human transmissions have also been reported in several NiV outbreaks and are usually associated with contact with Nipah patients’ secretions [21]. Human-to-human transmission occurs particularly when there is more than 12h exposure to body secretions of infected patients [22]. In fact, out of a total of 248 NiV infections reported in Bangladesh from 2001–2014, 84 cases, which is equivalent to approximately one third of cases, occurred as a result of person-to-person transmission [23]. In the Philippines, the source of human infection was traced to the consumption of horse meat or contact with infected horses, and then human-to-human transmission was reported from infected patients to healthy individuals [24]. The transmission cycle for NiV is presented in Figure 2.

## 4. The Virus: Unique Features of Henipa Virions

The Nipah virus (NiV) and Hendra virus (HeV) belong to the genus Henipavirus within the family *Paramyxoviridae*. The family includes a range of human and animal pathogens with potential to cause serious clinical outcomes. The family *Paramyxoviridae* has been categorized into genera based on the genome organization, sequence similarity, and biological activities of the encoded proteins as well as the virion characteristics. There are seven genera in family *Paramyxoviridae*, viz. Rubulavirus, Respirovirus, Morbillivirus, Avulavirus, Henipavirus, Aquaparamyxovirus, and Ferlavirus [25]. The major pathogens in the family *Paramyxoviridae* include the Measles virus, canine distemper virus, PPR (Peste-des-petits ruminants) virus belonging to the genus Morbillivirus; mumps virus (MuV), and human parainfluenza virus types 1–4 (hPIV1-4) belonging to the genus Rubulavirus; Newcastle disease virus belonging to the genus Avulavirus; and the deadly zoonotic Nipah virus (NiV) and Hendra viruses (HeV) belonging to the genus Henipavirus.

Several genetic attributes of NiV and HeVsetthem apart from other paramyxoviruses. First, these viruses have unique 3′ leader sequences that function as promoters for their transcription and unique 5′ trailer sequences which act as promoters from replication [26,27]. Second, in the transcriptase catalytic site, there is the presence of a highly conserved GDNE sequence instead of GDNQ (present in almost all of the non-segmented RNA viruses of negative polarity [27,28]. Third, the genome of henipaviruses is longer than viruses in other genera of the *Paramyxoviridae* family. The genome length of NiV is around 2700 nucleotides, which is 15% longer than other paramyxoviruses, while HeV’s genome length is 18,234 nucleotides [27,28]. However, J viruses of the family *Paramyxoviridae* have 700-nucleotide-longer genomes than henipaviruses. Yet, henipaviruses are unique in that they have long, untranslated regions present mostly at the 3′ end of all the transcription units except the L gene [27].

### 4.1. Structure and Genome Organization

Paramyxoviruses, including henipa virions, are pleomorphic; typically spherical (150 to 300 nm in diameter), although filamentous, forms that can reach up to 10 µmin length have been observed. The virions are enveloped with an envelope derived from the host cell plasma membrane. Two different viral transmembrane glycoproteins project from the surface of viral envelope: the attachment proteins termed G (for glycoprotein) and the fusion (F) proteins. The attachment proteins of henipaviruses are called G proteins in contrast to other paramyxoviruses (where they are called H or HN proteins) as the G proteins lack both haemagglutination and neuraminidase activities. These glycoproteins are packed densely on the viral envelope and are seen as a spike layer under cryo-electron microscopy [29]. The attachment (G) protein is responsible for receptor binding, while the F protein mediates the fusion of the viral envelope with target cell membranes. Paramyxoviruses have a non-segmented, (−) sense RNA genome, which is encapsidated by the nucleoprotein to form a flexible, loosely coiled structure named RNP (ribonucleoprotein) complex. The RNP complex serves as a template for viral RNA-dependent RNA polymerase (RdRp), which itself is composed of P (phosphoprotein) and L (large) proteins. The RNP complex is linked to viral membrane glycoproteins via matrix (M) protein, which plays a major role in particle assembly due to (i) its ability to act as a bridge between the RNP complex and envelope glycoproteins, (ii) its ability bind to the cellular membrane and cellular factors, and (iii) its property to self-assemble into higher-order oligomers [30].

The Henipavirus genome consists of single-stranded RNA of negative polarity, i.e., runningfrom 3′ to 5′. The genome consists of genes organized as N-P/V/C-M-F-G-L. The nucleocapsid protein (N) produced from the N gene is the most abundant protein and is responsible for the encapsidation of viral (−) sense RNA genome. N protein performs a wide range of functions, i.e., protects the viral RNA genome from degradation, prevents the viral RNA genome from being recognized by the host immune system, interacts with viral Matrix protein during virus assembly, and serves as a platform for viral genome replication [31].

The P gene encodes phosphoprotein (P), which, along with large (L) protein (encoded by L gene), forms the viral RNA-dependent RNA polymerase. It is the L protein which carries the catalytic activity of polymerase; however, interaction with the P protein is essential for binding to viral RNPs (ribonucleo proteins). The P gene also encodes accessory proteins, viz. V, and W proteins, by RNA editing(non-templated addition of Guanosine residues). The V protein is produced by the non-templated addition of 1G residue, while the W protein is produced by the non-templated addition of 2G residues. The V protein of henipaviruses has been studied in detail and has been found to bind to STAT (Signal Transducer and Activator of Transcription) proteins and prevent them from being imported to the nucleus, but does not induce the degradation of STAT proteins [32]. This is in contrast to rubulaviruses of the same family (*Paramyxoviridae*),which form a complex with STAT proteins and Cullin family ubiquitin ligases and result in polyubiquitination of STAT proteins, ultimately targeting them for degradation in the proteasome [33]. Other accessory proteins of henipaviruses are the C proteins, whichare produced from paramyxovirus P, V, and W genes by the use of alternate start codons (leaky scanning). Although the exact function of different C proteins is largely undetermined, they have been proposed to block interferon response, regulate transcription, and enhance virus budding via the recruitment of host factors. It is highly pertinent to mention that Cedar virus, a non-pathogenic henipavirus, lacks these proteins owing to the absence of RNA editing; thus, a huge immune response is evoked in human cells in response to Cedar virus infection.

The M gene encodes matrix (M) protein, which has been found to be a major coordinator of virus assembly and budding from the host cell plasma membrane. M proteins link the cytoplasmic tail of viral glycoproteins with the RNP complex and facilitate virus assembly. The fusion (F) protein is produced from the F gene. The F protein is a viral transmembrane glycoprotein (embeds in the viral envelope) and is responsible for the fusion of viral envelope with the plasma membrane of the host cell at neutral pH. The second glycoprotein is the Glyco- (G) protein, which is a viral attachment protein and recognizes and binds to corresponding receptors present on the host cell surface, viz. ephrin B2/B3 as receptors [33]. Neutralizing antibodies are directed to henipavirus G and F glycoproteins. The genome organization of henipa virions is presented in Figure 3.

### 4.2. Digging Deep into Henipavirus Replication Cycle

The first step in the henipavirus virus life cycle is the viral attachment to the target cell, which is followed by the fusion of the viral membrane to the host cell plasma membrane. Two major glycoproteins facilitate these events, viz. the viral attachment protein (G) and viral fusion (F) protein. The henipavirus attachment proteins, viz. G proteins, mediate binding to ephrin B2/3 receptors [34]. This binding results in a conformational change in the F protein, ultimately resulting in the activation of F andthus leading to the fusion of the viral membrane with the host cell plasma membrane at neutral pH, resulting in the delivery of the viral genome directly to the cell cytoplasm [35]. However, fusion cannot occur until viral F0 is cleaved into F1 and F2 subunits. For the majority of paramyxoviruses, the cleavage of F0 into F1 and F2 occurs at the multibasic residues by cellular proteases (furin) during trafficking through the trans-Golgi network [36]. However, for henipavirus F proteins, the presence of basic residues is not required for the cleavage of F0 into F1 and F2. Instead, F0 is transported to the cell surface; following this, it is endocytosed again and within the endosome, F0 is cleaved by cathepsin-L and trafficked back to the cell surface [37,38]. The F1 and F2 subunits produced by F0 cleavage are held together by disulfide linkages and penetrate the host cell plasma membrane following virion attachment to allow the fusion of viral and host plasma membranes. 

Once inside the cell cytoplasm, viral RNA genome undergoes active transcription and replication. During the initial stage of the viral life cycle, viral genomic RNA undergoes primary transcription to form mRNAs. During transcription, viral RNA-dependent RNA polymerase (RdRp) binds to 3′ end and synthesizes RNA; however, upon encountering intergenic regions (cis-acting elements), RdRp may fall off and thus fail to reinitiate the transcription of the next gene. This process is referred to as ‘start–stop’ transcription and results in the production of a gradient of mRNAs [31]. The mRNAs from the gene closest to the 3′ end of the viral genome are produced at the highest levels, while those from the gene closest to the 5′ end are produced at the lowest levels. The mRNAs thus produced are then capped and polyadenylated by viral L protein so that they can be recognized and translated by the host translational machinery. Once sufficient levels of viral proteins have accumulated, it triggers the switch from transcription to replication mode and the production of (+) sense antigenomes ensues [39]. Antigenomes further function as templates for the synthesis of (−) sense progeny genomes and the newly generated progeny genomes act as templatesfor secondary transcription, replication, and genomes of budding virions [39]. Once all the viral components (proteins and genome) are synthesized, they assemble together on the plasma membrane to allow budding of the progeny virions. An interesting feature of henipavirus budding is that F0, once transported to the cell membrane, are again internalized and cleaved into F1 and F2 subunits by endosomal cathepsin-L. The cleaved F protein (F1–F2 subunits) are present in the budding virions (Figure 4).

## 5. Traditional and Novel Diagnostic Tests

The diagnosis of NiV infections is made mainly on the basis of clinical symptoms, which can be challenging given the wide range of symptoms that are reported, the late onset of many of them, and the possibility that some individuals may be asymptomatic. Some of the techniques used for the diagnosis of NiV infections include PCR, ELISA, Virus neutralization, and cell fusion assays.

PCR followed by sequencing has been commonly employed for the diagnosis of NiV infections. The first NiV Real-time RT-PCR assays were created in 2004 using the N gene sequence. This PCR assay was found to be highly specific and could detect the viral RNA in blood samples from hamsters infected with NiV. However, this assay could not detect HeV RNA [40]. Zoologix recently created a kit based on this method that is able to detect NiV in different samples including blood, plasma, serum, cerebrospinal fluid, and tissues [41]. PCR followed by sequencing is essential for monitoring viral activity. For instance, a nested RT-PCR revealed the presence of two different NiV strains circulating in Thailand’s bat populations. Additionally, a real-time TaqMan array card that can concurrently detect 26 species linked to acute febrile sickness, including NiV, has been developed and compared to a typical real-time PCR (with a sensitivity of 88% and specificity of 99%).This assay may be helpful in outbreak circumstances for quick examination of a range of samples for a wide variety of pathogenic agents [42]. PCR assay targeting NiV N gene has also been employed for the generation of data for phylogenetic analysis [43].

Virus neutralization assays, which are considered as the standard for serology testing, were created shortly after the initial epidemic in Malaysia. Typically, Vero target cells are utilized in conventional NiV virus neutralization tests (VNTs). In this assay, the ability of sera to reverse viral cytopathic effects is taken into account and the sera able to do so are considered to be positively neutralizing. As NiV is a BSL-4 agent, to reduce the need for high containment labs for live NiV handling, multiple viruses have been pseudo typed to contain NiV glycoproteins, viz. F or G proteins [44]. One such example is a recombinant vesicular stomatitis virus (VSV) that carries the NiV glycoproteins.

NiV antigen detection and antibody response evaluation have both been done using Enzyme-Linked Immunosorbent Assays (ELISAs). One method for detecting viruses and differentiating between NiV and HeV is a monoclonal antibody-based N protein-capture ELISA [45]. Additionally, an antigen capture sandwich ELISA has been created using polyclonal antibodies obtained from rabbits immunized with the NiV G protein. Indirect ELISAs made with viral antigens used in field investigations on bats and other animals have been used to demonstrate seroconversion in humans. It has been demonstrated that human sera from NiV patients responds to recombinant NiV N, F, and G proteins after being successfully expressed in *E. coli* [46]. Additionally, a new IgG indirect ELISA has been developed that can identify antibodies in pig sera against full-length sub-cloned N proteins generated in *E. coli* and soluble HeV G or NiV G proteins expressed in *L. tarentolae* [47].

A measurable cell fusion assay utilizing reporter-gene activation has been reported for NiV. Target cells encoding a T7 promoter-driven *E. coli* lacZ cassette are co-cultured with effector cells thatexpress NiV F and G as well as a T7 RNA polymerase. NiV-mediated cell fusion can be quantified by measuring beta galactosidase activity when the cytoplasm of merged cells combines [47].

In new instances or in cases of NiV outbreaks, virus isolation is typically attempted using samples from the brain, lung, kidney, and/or spleen of infected individuals. NiV has been shown to grow well in Vero cells, and after three days of culture, a cytopathic effect is typically evident in the form of syncytia and distinctive plaques on the cell monolayer.

When accessible, electron microscopy and immuno-electron microscopy are additional helpful techniques for determining the unique structure of NiV and identifying viral antibody interactions, respectively [48,49]. Immunohistochemistry can be used to identify viral antigen in formalin-fixed tissues using anti-NiV antibodies [49].

With the aid of these methods and due to the characterization of NiV sequences after outbreaks, we now have a better understanding of the structure, divergence, and other facets of the virus’s life cycle. In particular, the development of low-containment diagnostic methods to quickly and accurately screen for non-neutralizing and nAbs that might be used in NiV endemic locations has been greatly aided by the use of recombinant viral proteins in place of natural virus.

## 6. Recent Strategies for the Control of the Nipah and Hendra Viruses

### 6.1. Passive Immunization Using Monoclonal Antibodies

Using recombinant antibody technology, a human monoclonal antibody has been generated that is specific for henipavirus G protein (m102.4). The m102.4 binds to ephrin B2/B3 binding sites on the G protein and thus blocks viral infection. The antibody can neutralize HeV, NiV-M (Malaysian strain), and NiV-B (Bangladesh strain) and can provide protection in ferrets and non-human primates when inoculated with a lethal dose of NiV (NiV-M and NiV-B) at various time points after exposure within a short therapeutic window [50,51]. The immunization with m102.4 would be most suited for post-exposure treatment in case of any future outbreak situations. To date, m102.4 has been administered to 14 individuals exposed to HeV in Australia (13 individuals) and NiV in the United States (1 individual) and no adverse effects have been noted in any of the individuals. m102.4 antibody was used in a randomized and controlled phase 1 study in Australia [52]. In this study, the antibody was found to be safe with no immunogenicity and a have half-life ranging from approximately 16.5 to 17 days.

Another monoclonal antibody that has been generated for henipaviruses is 5B3 (h5B3.1). The antibody is cross-reactive to NiV and HeV F proteins and binds to F protein on a pre-fusion conformation epitope, thereby preventing membrane fusion [53]. The antibody provides protection to NiV and HeV in ferrets when administered one to several days post challenge. Taken together, these studies show that passive immunization can be used to provide therapeutic benefit, allowing the infected individuals an extended period to mount a protective immune response [54].

### 6.2. Recent Developments in Vaccine Production

Since the G (attachment protein) and F (fusion protein) proteins of paramyxoviruses are the viral proteins against which neutralizing antibodies are directed to, NiV and HeV G and F proteins have been the targets for vaccination strategies.

#### 6.2.1. Subunit Vaccines

A soluble oligomeric form of the HeV G protein (HeV-sG) has been used as a subunit vaccine and has been found to protect animals including cats, ferrets, and African green monkeys against HeV and NiV challenge without any clinical signs of the disease, viral pathology, or even virus replication. The vaccinated animals developed high levels of neutralizing antibodies. The subunit vaccine has been approved in Australia for use in equines under the trade name of Equivac^®^ [55]. However, the vaccine resulted in the generation of low levels of neutralizing antibodies in pigs following NiV challenge [56].

#### 6.2.2. Vectored Vaccines

Various viral vectors have been used for the generation of NiV vaccine candidates including pox viruses, adeno-associated viruses, rhabdoviruses, and paramyxoviruses.

Poxviruses including the vaccinia virus that expresses NiV for G proteins have been produced and used in experiments to vaccinate a hamster model. These proteins, irrespective of being expressed alone or together, were able to provide complete protection against NiV challenge. Passive antibody transfer from vaccinated hamsters (exhibiting a high titer of neutralizing antibodies) to naïve hamsters was also observed [40]. A canarypox vaccine vector that expresses NiV F and G proteins either alone or in combination has been recently developed. This vaccine has been used in experiments to immunize pigs and was found to result in the production of neutralizing antibodies without virus shedding or the exhibition of histopathological lesions. The simultaneous expression of both of NiV surface glycoproteins, viz. F and G proteins, was found to produce the highest neutralizing antibody responses [57].

Adeno virus and Adeno-associated virus (AAV) have also been used as vectors for NiV vaccine production. Adeno virus belongs to the *Adenoviridae* family; however, Adeno-associated virus (AAV) belongs to the *Parvoviridae* family. AAV-expressing NiV (Malaysian strain) G protein has been produced and provided complete resistance to NiV-M challenge in hamsters with no symptoms of clinical illness [58]. Human adenovirus (HAV) vectors possess a problem with preexisting immunity; however, chimpanzee adenoviral (ChAd) vectors do not have such issues [59]. Previously, Oxford 1 (ChAdOx1), a genetically modified chimpanzee adenoviral vector, was investigated for use in the generation of a NiV/HeV vaccine [60]. The ChAdOx1-expressing G glycoprotein from NiV-B (ChAdOx1 NiV-B) has been used to immunize hamsters via the intramuscular route, either as a single shot or as part of a prime-boost strategy followed by virus challenge 42 days after the booster or the single vaccine with NiV-B. Only one dose of ChAdOx1 expressing NiV-B G protein was shown to be adequate to provide complete protection against NiV-B.

Rhabdoviruses, including VSV and rabies virus, have also been utilized in an attempt to develop a NiV vaccine. As vaccine candidates, VSV vector platforms have been exploited in a variety of ways. These include using replication-competent VSV that expresses NiV F, G, or N proteins [61]. The VSV vaccines expressing G from NiV-M and NiV-B strains were sufficient to protect both Syrian hamsters and ferrets in a single dose. Animals vaccinated with NiV F- or G-expressing vectors exhibited high neutralizing Ab titers, low viral RNA/antigen, and exhibited no pathological alterations [62,63]. Only partial protection was conferred in hamsters that had been vaccinated with recombinant VSV-expressing N protein from NiV, implying the possibility that both cellular and non-neutralizing antibody responses contribute to protection against disease caused by NiV [64].

Evaluations have also been conducted on rabies virus SAD B19 vaccine strain that expresses HeV or NiV G proteins [65]. In mice, this recombinant virus was utilized to express the wild-type or codon-optimized version of the HeV G gene [61]. The F or G protein-expressing recombinant rabies virus(Evelyn–Rokitnicki–Abelseth strain) was recently tested in mice and pigs [66]. This vector can also serve as an oral vaccine in dogs. This live-attenuated vector, while not suited for human use, is a potential veterinary vaccine for NiV because it is already authorized for use in some species and may be modified for use in emergency situations to guard against NiV infection in livestock, particularly swine.

Paramyxoviruses including recombinant Measles virus and recombinant Newcastle disease virus have also been used as vaccine vectors. The recombinant measles virus expressing NiV glycoprotein provided complete protection against NiV challenge in hamsters [67]. In addition, a paramyxovirus a recombinant Newcastle disease virus that expresses the NiV G or F proteins has also been described [68]. NDV vectors are better vaccine platforms, as they can be propagated in chicken eggs in high titers without the need to use cell culture.

Other viruses, such as the Venezuelan equine encephalitis virus encoding NiV F or G proteins, have been used as vaccine candidates and have shown highly potent neutralizing antibody responses in mice [47].

### 6.3. Virus-like Particles

Virus-like particles (VLPs) are nanostructures thatclosely resemble viruses but do not contain viral genome and are therefore non-infectious. The expression of selected viral proteins via expression plasmids results in the spontaneous assembly and release of VLPs. Since these particles possess many characteristics of authentic virus particles, such surface dimensions and structure, they are very efficient immunogens. Furthermore, these particles are very safe owing to the absence of viral genetic material. Vaccines based on the VLP platform that have been approved for use in humans against viruses include Gardasil against human papillomavirus and Sci-B-Vac and Bio-HepB against Hepatitis B virus. VLP-based vaccines are being developed for many non-enveloped and enveloped viruses, including NiV. For the production of NiV VLPs, three NiV proteins, namely, F, G, and M proteins, have been shown to be crucial. These three proteins have been expressed simultaneously in mammalian cells for NiV VLP production. The VLPs produced by the simultaneous expression of F, G, and M proteins have been found to be fusogenic and to induce syncytia formation. The VLPs have been found to be composed of all the three expressed proteins with intracellular processing similar to that observed for NiV virions. Moreover, the VLP morphology was similar to the NiV virions. Furthermore, these VLPs activated the innate immune system, as evidenced by PCR array analysis [69]. These VLPs have also been evaluated in a Balb/c mouse model and have been proven to generate neutralizing antibodies, as well as to provide protection to hamsters against NiV challenge following the administration of three doses of the vaccine and also with a single-shot [70].

### 6.4. Single-Dose Lipid Nanoparticle mRNA Vaccine

Because of their safety, efficacy, and ease of use, messenger RNA (mRNA)-based vaccines have become a popular vaccine technique lately. The limited efficiency of HeV-sG mRNA lipid nanoparticle has been reported. Though encouraging, there exists the need for further adjuvant addition, adjuvant route modification, and/or prime-boost vaccination strategies. Recently, the National Institute of Allergy and Infectious Diseases (USA) launched a phase 1 clinical trial to evaluate an experimental vaccine based on an mRNA platform. The vaccine, mRNA-1215 Nipah vaccine, was developed by Moderna (Cambridge, MA, USA) and involves the same technology as was used in many approved COVID-19 vaccines. The phase 1 trial is being carried out in Maryland at the NIH Clinical Centre, Bethesda [71].

## 7. Other Recently Discovered Henipaviruses

Apart from the Nipah and Hendra viruses, other henipaviruses have now been discovered, namely, Cedar virus, Ghana virus, and Mojaing virus. Cedar virus (CedPV), a non-pathogenic member of the genus henipavirus, was isolated from urine samples collected from fruit bats in Australia [72]. CedPV lacks the RNA editing site in its phosphoprotein gene (P), commonly found in other paramyxovirus genomes, and thus results in a lack of expression of accessory proteins, i.e., V and W proteins. Since V and W proteins are interferon antagonists, CedPV is not able to counter the antiviral defense mounted by the host cells and as such CedPV infection does not result in clinical disease in ferrets and guinea pigs (lab animals commonly used to study henipavirus pathogenesis). The Ghana virus (also called Kumasi virus) RNA was first detected in 2009 from straw-colored fruit bat (*Eidolon helvum*) [73]. The zoonotic potential of the Ghana virus (GhV) is unknown, as no cases have been reported in humans or livestock. The Mojiang virus is a rat-borne henipavirus isolated from Chinese miners exhibiting symptoms of lethal pneumonia in 2012. The Mojiang virus is closely related to bat-borne henipaviruses; however, it lacks the conserved ephrin receptor-binding motif in the attachment protein and does not interact with known paramyxovirus receptors, including sialic acid, ephrin B2/B3, and CD150 [74].

## 8. Conclusions and Future Perspectives

The Nipah and Hendra viruses are highly virulent zoonotic paramyxoviruses with a case fatality rate of up to 75%. Since the initial outbreak of the Nipah virus in Malaysia in 1998, small numerous outbreaks have occurred in Southeast Asia, particularly in India and Bangladesh, where cases are reported almost every year, leading to a death toll of 260 people worldwide todate [4]. AsNipah and Hendra viruses are zoonotic BSL-4 viruses for which no vaccines or antivirals are currently available, symptomatic treatment becomes the only option available for Nipahvirus-affected individuals. Recent studies have shown that the soluble oligomeric form of HeV G protein, viz. HeV-sG vaccine, has the ability to protect African green monkeys against Nipah virus disease for seven days following immunization [75]. Recently, Oxford University and the National Institute of Health conducted conducted preliminary trials using chimpanzee adenovirus vector-expressing NiV glycoprotein and found that the vaccine was able to provide near-complete protective immunity against NiV in African green monkeys [6]. Such studies could pave the way for development of a vaccine that could provide protection against the Nipah and Hendra viruses; in the absence of which, these viruses pose have the potentialto cause the next pandemic, which can be anticipated to have a heavy toll on human life as the case fatality rate of the Nipah virus is very high. The One Health strategy to immunization could be beneficial for preventing and controlling NiV disease in domestic animals and limiting its spread to people. The effectiveness of the licencedEquivac^®^HeV vaccination in protecting horses from the virus is a very pertinent illustration of how this might be possible.

## Figures and Tables

**Figure 1 pathogens-11-01419-f001:**
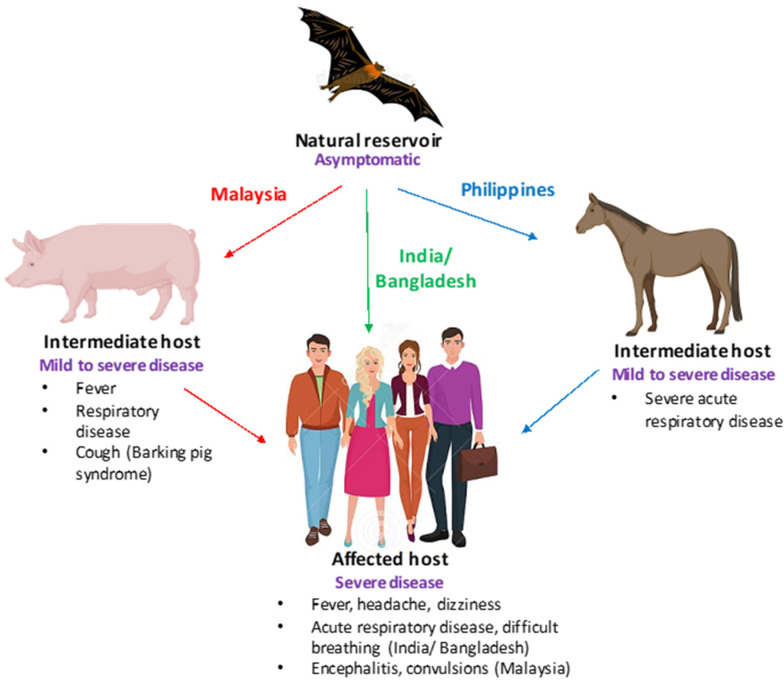
Clinical presentation of Nipah virus (NiV) infections. A schematic depicting the clinical presentation of NiV infection is shown. Bats (flying foxes) are the natural reservoirs of NiV and are asymptomatic. In swine (Malaysia) and equine (Philippines),NiV infection is characterized by severe respiratory disease, cough, and neurological signs, while in humans fever, encephalitis (Malaysia), and acute respiratory disease (Bangladesh) have been reported.

**Figure 2 pathogens-11-01419-f002:**
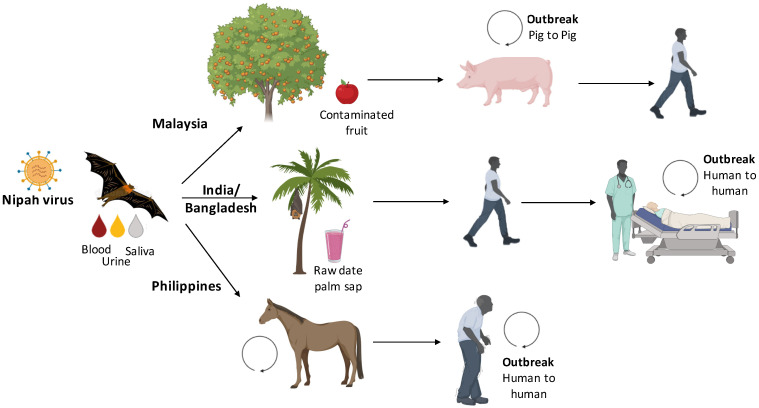
Schematic representation of different Nipah virus (NiV) transmission routes. In Malaysia, pig farms lie in close proximity to fruit trees, which are resided by fruit bats and domestic pigs contract infection by coming into contact with materials contaminated by bats, particularly by consumption of bat-eaten fruits or by exposure to bat urine. NiV is subsequently transmitted from pigs to humans by direct contact. In India and Bangladesh, the major route of NiV transmission from bats to humans is by the consumption of raw date palm sap contaminated with bat saliva or urine followed by person-to person transmission occurring by close contact. In the Philippines, the source of human infection was traced to the consumption of horse meat or contact with infected horses, and then human-to-human transmission was reported from infected patients to healthy individuals.

**Figure 3 pathogens-11-01419-f003:**
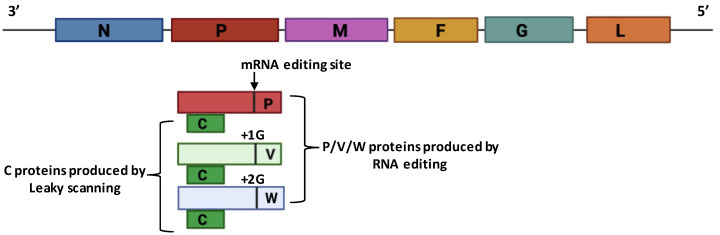
Genome organization of henipa virions. A schematic depicting the gene order in henipaviruses is shown. Henipaviruses produce P, V, and W proteins as a result of mRNA editing (non-templated addition of Guanosine residues), while C protein is produced by leaky scanning (use of alternative start codons).

**Figure 4 pathogens-11-01419-f004:**
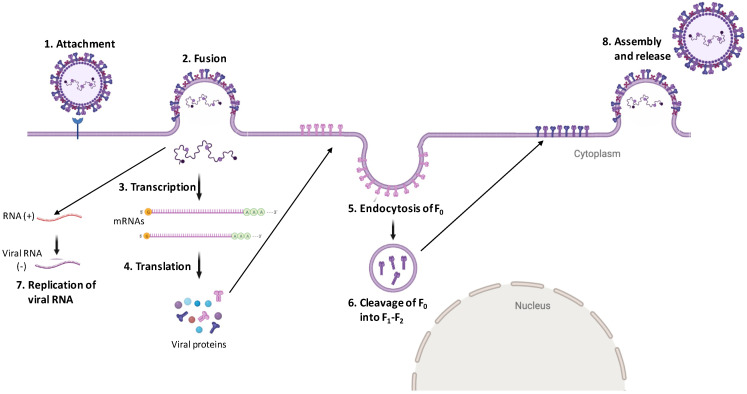
Replication cycle of henipaviruses: The viruses attach via G glycoproteins (shown in blue) to cellular receptors (ephrin B2/3). This triggers a conformational change in viral F proteins (shown in purple color), which allows the fusion of the viral envelope with the host cell plasma membrane, resulting in release of viral (−) RNA directly into the cell cytoplasm. This is followed by transcription to form mRNA followed by translation to allowthe formation of viral proteins. The F protein is produced in an inactive F0 precursor (pink), which is trafficked to the cell membrane from whichit is endocytosed and cleaved into active F1–F2 form by the cathepsin proteases present in the endosomes. The cleaved F proteins along with other viral proteins then assemble at the cell surface to allow budding and the release of henipa virions.

## Data Availability

Not applicable.

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
