# Peer review of "Nipah and Hendra Viruses: Deadly Zoonotic Paramyxoviruses with the Potential to Cause the Next Pandemic"

_pathogens, 2022, doi:10.3390/pathogens11121419_

Round 1

Reviewer 1 Report

  • Minor
        • There are many typos (below are some of them). Please fix

    Line 33: isolatedfrom > isolated from
    Line 38: presentationsranging > presentation ranging
    Line 48: conductresearch >  conduct research
    Line 208: 10um > 10μm (10 μm? Check the journal recommendations)
    Line 310: includingblood > including blood
    Line 353: methodsand > methods and

        • Reference style inconsistent: ..analysis [39]  vs proteins[40] Check journal style.

        • Line 339. More than one cell fusion assay has been developed. Mention it, it would be appropriate to include the references as well.

        • Abuse of “viz.”

    Major
        • Section 1-4. The information included in these sections has been covered in many reviews. It seems irrelevant at this point. There is no point in including it.

        • Section 5. It is a nice touch to focus on the unique features of these viruses please expand the section with this idea in mind. Otherwise, the section feels, once again, irrelevant.

        • Sections 7.3 and 7.4 receive little attention in comparison to 7.2. Please expand.

Author Response

We have incorporated the suggestions raised by honorable reviewers in true spirit. Authors feel that the suggestions raised by honorable reviewers have enhanced the readability of the current manuscript. The text has been edited using track changes and therefore colored red in the manuscript.  The point-wise justifications of the revision made in the manuscript are as follows:

Reviewer #1

Comments and Suggestions for Authors

Minor
    • There are many typos (below are some of them). Please fix
Line 33: isolatedfrom > isolated from
Line 38: presentationsranging > presentation ranging
Line 48: conductresearch >  conduct research
Line 208: 10um > 10μm (10 μm? Check the journal recommendations)
Line 310: includingblood > including blood
Line 353: methodsand > methods and

Response: All typos have been corrected in the revised manuscript

  • Reference style inconsistent: ..analysis [39]  vs proteins[40] Check journal style.

Response: Corrected in the revised manuscript

  • Line 339. More than one cell fusion assay has been developed. Mention it, it would be appropriate to include the references as well.

Response: Since cell fusion assay is not the focus of this review, we have just introduced it in the manuscript.

  • Abuse of “viz.”

Response: We have reduce the use of “viz.”

Major
    • Section 1-4. The information included in these sections has been covered in many reviews. It seems irrelevant at this point. There is no point in including it.

Response: Section 3 has been merged with section 2 after modifications.

  • Section 5. It is a nice touch to focus on the unique features of these viruses please expand the section with this idea in mind. Otherwise, the section feels, once again, irrelevant.

Response: We have expanded this section by adding details about how V proteins of henipaviruses function differently from other paramyxoviruses (do not induce degradation of STAT proteins)

  • Sections 7.3 and 7.4 receive little attention in comparison to 7.2. Please expand.

Response: We have expanded the sections 7.3 and 7.4. We have also included the recent ongoing clinical trial on mRNA based Nipah virus vaccine.

Authors are grateful to the honorable reviewers for helping in improving the manuscript.

Reviewer 2 Report

Title: Nipah and Hendra Viruses: Deadly Zoonotic Paramyxoviruses with Potential to Cause the Next Pandemic

In this review article, the authors descript Nipah and Hendra viruses are deadly zoonotic paramyxoviruses with a related aspects of 1) Nipah virus: a cause of grave concern; 2) Disease patterns: Past outbreaks and current scenario; 3) Transmission cycle from bats to humans; 4) The virus: Unique features of henipa virions; 5) Traditional and novel diagnostic tests, and Recent strategies for control of Nipah and Hendra viruses et al..

 Major comments:

1. Some excellent Henipavirus (NiV and HeV) reviews have been published before and recently (listed below). The current review content is very similar to these. I suggest that authors should focus on a specific topic and cite recent research reports. There are more than 10 publications related to NiV and HeV in 2022.

2. References are not cited properly in this review. For example, in the Introduction, the authors provide a lot of background information on Henipavirus and other WHO research activities, citing only one website (WHO). Authors need to cite original and recent research papers that contain these points to support the statement.

Minor comments:

1.           Line 18: “upto” should be changed to up to

2.           Section 2 only discussed NiV. The title of this review is Nipah and Hendra viruses. Please provide the necessary introduction for HeV.

3.           Figure 1, No HeV Clinical presentation.

4.           Some background information about HeV in section 3 should be combined with section 2.

5.           Figure 2 demonstrated only NiV transmission route without showing HeV.

6.           In section 5.2, A figure needs to be provided to show Henipavirus replication cycle.

7.           Section 6, please have a table summarising the diagnostic tests.

8.           7.1 Passive immunization using m102.4 should be changed to Passive immunization using monoclonal antibodies, since you talked about another mAb (h5B3.1) in this section.

9.           Line 353: “methodsand” should be changed to methods and.

10.         Line 466: The limited efficiency of HeV- 466 sG mRNA lipid nanoparticle was reported in the investigations.

11.         Line: 502: The reference “Thakur et al., 2022” was not correctly formatted.

*This review is valuable and well written. However, it needs to focus on specific topics with recent discoveries. Therefore, it needs major revision before being published in a peer-reviewed journal.

References:

Nipah virus: a recently emergent deadly paramyxovirus by KB Chua, WJ Bellini, PA Rota, BH Harcourt, A Tamin, Science, 2000 – science.org.

Hendra and Nipah infection: emerging paramyxoviruses by M Aljofan - Virus research, 2013

Henipaviruses—A constant threat to livestock and humans by S Kummer, DC Kranz - PLoS Neglected Tropical Diseases, 2022.

Nipah virus: epidemiology, pathology, immunobiology and advances in diagnosis, vaccine designing and control strategies – a comprehensive review by Raj Kumar Singh, Kuldeep Dhama, Sandip Chakraborty, Ruchi Tiwari, Senthilkumar Natesan, Rekha Khandia, Ashok Munjal, Kranti Suresh Vora, Shyma K. Latheef, Kumaragurubaran Karthik, Yashpal Singh Malik, Rajendra Singh, Wanpen Chaicumpa & Devendra T. Mourya (2019), Veterinary Quarterly, 39:1, 26-55, DOI: 10.1080/01652176.2019.158082

Nipah virus by B Rathish, K Vaishnani - StatPearls [internet], 2022

Author Response

We have incorporated the suggestions raised by honorable reviewers in true spirit. Authors feel that the suggestions raised by honorable reviewers have enhanced the readability of the current manuscript. The text has been edited using track changes and therefore colored red in the manuscript.  The point-wise justifications of the revision made in the manuscript are as follows:

Reviewer #2

Title: Nipah and Hendra Viruses: Deadly Zoonotic Paramyxoviruses with Potential to Cause the Next Pandemic

In this review article, the authors descript Nipah and Hendra viruses are deadly zoonotic paramyxoviruses with a related aspects of 1) Nipah virus: a cause of grave concern; 2) Disease patterns: Past outbreaks and current scenario; 3) Transmission cycle from bats to humans; 4) The virus: Unique features of henipa virions; 5) Traditional and novel diagnostic tests, and Recent strategies for control of Nipah and Hendra viruses

 Major comments:

  1. Some excellent Henipavirus (NiV and HeV) reviews have been published before and recently (listed below). The current review content is very similar to these. I suggest that authors should focus on a specific topic and cite recent research reports. There are more than 10 publications related to NiV and HeV in 2022.

Response:  Section 3 has been merged with section 2 with some modiciations.

  1. References are not cited properly in this review. For example, in the Introduction, the authors provide a lot of background information on Henipavirus and other WHO research activities, citing only one website (WHO). Authors need to cite original and recent research papers that contain these points to support the statement.

Response: References have been cited in the revised manuscript (introduction section): Rogers et al., 1996 and Chua et al., 1999

Minor comments:

  1. Line 18: “upto” should be changed to up to

Response: Corrected in the revised manuscript

  1. Section 2 only discussed NiV. The title of this review is Nipah and Hendra viruses. Please provide the necessary introduction for HeV.

Response: HeV has been introduced in the section in the revised manuscript; section 3 merged with section 2

  1. Figure 1, No HeV Clinical presentation.

Response: HeV clinical presentation has been included in the text in section 2

  1. Some background information about HeV in section 3 should be combined with section 2

Response:  Section 3 merged with section 2

  1. Figure 2 demonstrated only NiV transmission route without showing HeV.

Response: The transmission route of HeV has been provided in the text

  1. In section 5.2, A figure needs to be provided to show Henipavirus replication cycle.

Response: A figure demonstrating henipavirus replication has been incorporated (Fig. 4)

  1. Section 6, please have a table summarising the diagnostic tests.

Response:  Details of all diagnostic tests have already been provided in text

  1. 7.1 Passive immunization using m102.4 should be changed to Passive immunization using monoclonal antibodies, since you talked about another mAb (h5B3.1) in this section.

Response: Done

  1. Line 353: “methodsand” should be changed to methods and.

Response: Corrected in the revised manuscript

  1. Line 466: The limited efficiency of HeV- 466 sG mRNA lipid nanoparticle was reported in the investigations.

Response: Corrected

  1. Line: 502: The reference “Thakur et al., 2022” was not correctly formatted.

Response: Formatted in the revised version

Authors are grateful to the honorable reviewers for helping in improving the manuscript.

With deep regards

Round 2

Reviewer 1 Report

.

Reviewer 2 Report

It is ok.